# Symmetry breaking in reconstituted actin cortices

**Enas Abu Shah[1,2], Kinneret Keren[1,2,3]\***

[1]Department of Physics, Technion–Israel Institute of Technology, Haifa, Israel; [2]The Russell Berrie Nanotechnology Institute, Technion–Israel Institute of Technology, Haifa, Israel; [3]Network Biology Research Laboratories, Technion–Israel Institute of Technology, Haifa, Israel

**Abstract** The actin cortex plays a pivotal role in cell division, in generating and maintaining cell polarity and in motility. In all these contexts, the cortical network has to break symmetry to generate polar cytoskeletal dynamics. Despite extensive research, the mechanisms responsible for regulating cortical dynamics in vivo and inducing symmetry breaking are still unclear. Here we introduce a reconstituted system that self-organizes into dynamic actin cortices at the inner interface of water-in-oil emulsions. This artificial system undergoes spontaneous symmetry breaking, driven by myosin-induced cortical actin flows, which appears remarkably similar to the initial polarization of the embryo in many species. Our in vitro model system recapitulates the rich dynamics of actin cortices in vivo, revealing the basic biophysical and biochemical requirements for cortex formation and symmetry breaking. Moreover, this synthetic system paves the way for further exploration of artificial cells towards the realization of minimal model systems that can move and divide.

## Introduction

The actin cytoskeleton plays a central role in many cellular processes including polarization, cell shape determination, intracellular transport, cell division and movement (*Pollard and Cooper, 2009*). The structure and function of the cytoskeleton arise from the self-organized dynamics of numerous molecular building blocks. This self-organization spans several orders of magnitude in space and time and involves a complex interplay between biochemical and biophysical processes; A myriad of proteins interact with the actin cytoskeleton and influence its behavior, in a manner that is dependent on the global mechanical properties of the network but at the same time determines it (*Lecuit and Lenne, 2007*; *Pollard and Cooper, 2009*; *Mullins and Hansen, 2013*). Despite the significant progress in uncovering the molecular details underlying cytoskeletal dynamics, the principles governing large-scale coordination and polarization of the cytoskeleton are still not well-understood.

The realization of biomimetic systems that reconstitute cellular processes in vitro, detached from the complexity of the cell, is a powerful approach for dissecting complex cellular phenomena. In particular, in vitro experiments have significantly advanced our understanding of the molecular requirements and the biophysical principles underlying actin-based motility and cytoskeletal organization in bulk (*Welch et al., 1998*; *Cameron et al., 1999*; *Loisel et al., 1999*; *Gardel et al., 2004*; *Van Der Gucht et al., 2005*; *Bendix et al., 2008*; *Field et al., 2011*; *Kohler et al., 2012*), and more recently in cell-sized compartments (*Pontani et al., 2009*; *Stachowiak et al., 2009*; *Pinot et al., 2012*; *Sanchez et al., 2012*; *Carvalho, 2013b*). However, we are still far from understanding the complexity of cytoskeletal dynamics in vivo, and recapitulating even basic cellular phenomena such as polarization, division and directed movement in synthetic systems remains an outstanding challenge.

The actin cytoskeleton undergoes continuous turnover and remodeling which are essential for its ability to perform its cellular tasks (*Pollard and Cooper, 2009*). In particular, the thin cortical actin shell

**\*For correspondence:** kinneret@ph.technion.ac.il

**Competing interests:** The authors declare that no competing interests exist.

**eLife digest** Cells are extremely complex because they have to perform a vast number of processes. However, this also makes it difficult for researchers to figure out how the individual parts of the cell work. There is interest, therefore, in developing simple artificial cells that can accurately mimic how specific parts of a cell behave.

An important process for a cell is called polarization. This is where the contents of the cell arrange themselves in a way that is not symmetrical. Polarization is necessary for many cellular functions, and is particularly important during embryonic development where it helps to form the complex shape of the developing embryo.

The cytoskeleton—a dynamic structure that supports the cell and enables it to move—is crucial for polarization. An important part of the cytoskeleton is the actin cortex. This is a thin active sheet made up of a network of tiny filaments of a protein called actin that assembles at the inner face of the cell membrane. Many aspects of the structure and behavior of the actin cortex are not understood.

Abu Shah and Keren have now developed an artificial cell system using aqueous droplets surrounded by oil that can reproduce the behavior of actin cortices in real cells. An actin cortex forms upon the localization of specific nucleation factors at the inner surface of the droplets.

The artificial cortices are capable of spontaneous symmetry breaking, similar to the initial polarization in embryonic cells during development. This symmetry breaking is driven by molecular motors called myosins and depends on the connectivity of the actin network in the cortex. Experiments on the artificial cells also rule out several other mechanisms that have been proposed to explain symmetry breaking.

The work of Abu Shah and Keren represents a further step towards the goal of creating simple artificial cells that can move and divide.

underneath the cell membrane undergoes continuous assembly and disassembly processes, catalyzed by nucleation-promoting factors localized at the membrane and disassembly factors (*Fritzsche et al., 2013*). Among the nucleation-promoting factors, Arp2/3 which nucleates branched networks localizes to cortical actin networks (*Machesky et al., 1994*) and is essential for cortex formation (*Bovellan, 2012*). Formins, which nucleate linear filaments, were also found to localize to cortical actin networks, yet their role is still not entirely clear (*Bovellan, 2012*; *Fritzsche et al., 2013*). A host of actin binding proteins, including myosin motors, tethering proteins and various crosslinkers, further contribute to the spatio-temporal organization of actin cortices in cells (*Munro et al., 2004*). This dynamic remodeling is responsible for the global rearrangements of the actin cortex which are essential for its function during polarization and movement (*Salbreux et al., 2012*).

Polarization in cells typically occurs in response to internal or external cues. Yet, the onset of polarity often reflects an inherent instability mechanism which can lead to symmetry breaking in the absence of any directional cues (*Van Oudenaarden and Theriot, 1999*; *Verkhovsky et al., 1999*; *Wedlich-Soldner et al., 2003*; *Boukellal et al., 2004*; *Van Der Gucht et al., 2005*; *Carvalho et al., 2013a*, *2013b*). While biochemical signaling pathways appear to be important for transducing directional cues, the instability, at least in some cases, can be primarily mechanical (*Mullins, 2010*; *Van Der Gucht and Sykes, 2009*). For example, reconstitution of actin-based motility of bacterial pathogens such as *Listeria monocytogenes* revealed that directional movement can arise from spontaneous symmetry breaking within the actin network which ruptures and forms a polar comet-tail (*Cameron et al., 1999*; *Van Oudenaarden and Theriot, 1999*; *Boukellal et al., 2004*; *Van Der Gucht et al., 2005*). In the cell cortex, the uniform actin shell can break symmetry to form a polar network which is essential for generating and maintaining cell polarity (*Munro et al., 2004*; *Cowan and Hyman, 2007*; *Munro and Bowerman, 2009*; *Goehring et al., 2011*; *Salbreux et al., 2012*) and inducing directional cell movement (*Hawkins et al., 2011*; *Poincloux et al., 2011*). Despite their importance, the factors controlling symmetry breaking in the cell cortex remain poorly understood. An attractive hypothesis is that the instability leading to cortical symmetry breaking is also mechanical as in the case of comet tail motility (*Van Oudenaarden and Theriot, 1999*; *Van Der Gucht et al., 2005*; *Dayel et al., 2009*), yet the mechanisms involved appear different. Comet tail formation is a myosin–independent process

driven by actin polymerization (*Loisel et al., 1999*), whereas cortical symmetry breaking appears to rely on myosin as the main force generating element; myosin contraction which generates cortical actin flows has been shown to be essential for inducing polarization and defining the anterior–posterior axis during the early stages of embryogenesis in *Caenorhabditis elegans* and in other species (*Munro et al., 2004*; *Mullins, 2010*; *Munro and Bowerman, 2009*; *Mayer et al., 2010*).

Reconstitution of actomyosin networks is a valuable tool for dissecting the mechanisms underlying cortical symmetry breaking. Recent in vitro experiments revealed that small changes in the relative amounts of myosin and crosslinkers, or in their activity, can lead to a sharp transition in the overall contractile behavior of reconstituted actin networks (*Bendix et al., 2008*; *Kohler et al., 2012*). However, to emulate cortical dynamics it is essential to incorporate actin turnover dynamics, couple the actin network to a soft interface (*Murrell and Gardel, 2012*), and reproduce a cell-like geometry (*Pontani et al., 2009*; *Pinot et al., 2012*; *Carvalho et al., 2013b*). Recent reports (*Carvalho et al., 2013a*, *2013b*) suggest that myosin is responsible for building tension in actin networks tethered to soft interfaces, and can induce cortical rupture and generate asymmetry. Here we report the realization of artificial actin cortices that exhibit spontaneous symmetry breaking and display cell-like behavior. This novel synthetic system provides a basis for studying cortical actin dynamics and polarization in a simplified environment detached from the complexity of the living cell. Moreover, it presents an interesting example of a self-organized, far-from-equilibrium, active system that utilizes chemical energy to polarize and generate directional forces.

## Results

### Reconstitution of artificial actin cortices in cell-like compartments

The formation of artificial actin cortices was induced by localizing the ActA protein from the pathogenic bacteria *L. monocytogenes* to the inner interface of water-in-oil emulsions (*Figure 1*). ActA is a nucleation-promoting factor, known for its role in activating the Arp2/3 complex and inducing nucleation of branched actin networks (*Welch et al., 1998*). Such branched actin networks are found in the cortices of many cell types (*Machesky et al., 1994*; *Medalia et al., 2002*; *Bovellan, 2012*; *Fritzsche et al., 2013*). A soluble ActA construct was purified from *L. monocytogenes* and conjugated to a fluorescent hydrophobic linker made with Bodipy-FL, to generate an amphiphilic complex ('Materials and methods'). This amphiphilic ActA complex was mixed with diluted *Xenopus laevis* egg cytoplasmic extract supplemented with labeled actin, and encapsulated within droplets surrounded by mineral oil ('Materials and methods'). The amphiphilic ActA spontaneously localized to the water–oil interface following droplet formation. Upon localization, ActA induced the assembly of a cortical actin network at the inner surface of the aqueous droplets (*Figure 1A,D*). Cortex assembly initiated in patches scattered throughout the interface, consistent with the autocatalytic nature of Arp2/3 mediated actin filament nucleation (*Mullins and Hansen, 2013*; *Figure 1F*). The system reached a steady state after ~10 min, with the formation of thin (~1.5 µm; *Figure 1—figure supplement 1*), homogenous actin cortices. The assembled cortices are dynamic, as illustrated by the rapid cortical recovery following photobleaching (*Figure 1E*, *Figure 1—figure supplement 2*; *Video 1*). Thus, the cortical steady state reflects a balance between actin assembly and disassembly at the interface, as in cells (*Fritzsche et al., 2013*). Control experiments with soluble ActA, or in the absence of ActA, resulted in an essentially uniform actin distribution throughout the emulsion droplets, with no apparent cortical localization (*Figure 1B,C*), indicating that cortex formation depends on the localization of a nucleation-promoting factor at the interface.

### Temperature dependent symmetry breaking in reconstituted actin cortices

Our reconstituted actin cortices can exhibit spontaneous symmetry breaking (*Figure 2*). The actin network assembled within emulsion droplets incubated at 20°C broke symmetry and formed polar actin caps (*Figure 2A*), in contrast to the homogenous spherical cortices formed at higher temperature (30°C; *Figure 2B*). We followed the dynamics of actin cap formation as a function of time (*Figure 2C,D*; *Video 2*). Initially, actin assembly appeared nearly uniform throughout the interface, but within minutes we observed the onset of a directional actin network flow along the interface. The cap formation process developed over ~10–20 min, with cortical flows initiating at ~1–4 µm/min (*Figure 2E*), and slowing down over time, as the cap contracted (*Figure 2D*). The local turnover due to actin assembly

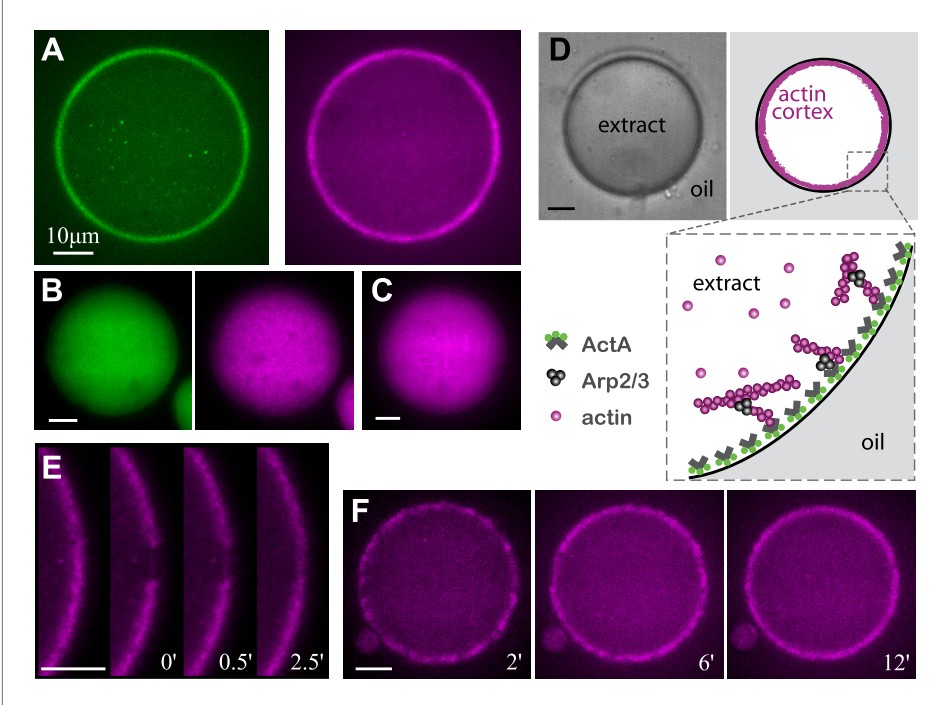

**Figure 1**. Reconstitution of actin cortices within water-in-oil emulsions. (**A**) Spinning disk confocal images of bodipy-conjugated ActA (left) and rhodamine-labeled actin (right) in a water-in-oil emulsion. The bodipy-conjugated ActA localizes to the water–oil interface, and induces the formation of an actin cortex there. The actin signal reflects the distribution of actin monomers and filaments. (**B**) Images of AlexaFluor488-conjugated ActA (left panel) and rhodamine-labeled actin (right) in a water-in-oil emulsion. The hydrophilic ActA and the actin remain dispersed throughout the emulsion. (**C**) Image of rhodamine-labeled actin in a water-in-oil emulsion in the absence of ActA. The actin is distributed within the emulsion. (**D**) Schematic illustration of the actin cortex formed at the inner interface of a water-in-oil emulsion. A bright-field image (top left) and a scheme (top right) of an actin cortex at the inner interface of an aqueous droplet surrounded by oil. The zoomed scheme (bottom) illustrates the localization of the amphiphilic bodipy-conjugated ActA to the water–oil interface, which leads to local activation of Arp2/3 and nucleation of a cortical actin network. (**E**) Scanning confocal images showing cortical recovery in a photobleaching experiment (***Video 1***). The time after photobleaching is indicated. (**F**) Spinning disk confocal images from a time-lapse video showing the formation of a homogenous actin cortex. The time after droplet formation is indicated. Scale bars: 10 μm.

The following figure supplements are available for figure 1:

**Figure supplement 1**. Actin and ActA distributions in water-in-oil emulsions.

**Figure supplement 2**. Analysis of photobleaching experiments.

**Figure supplement 3**. Actin cortices formed with different types of labeled actin.

and disassembly continued throughout the cap formation process, as illustrated by photobleaching experiments in which a cortical region within the asymmetric actin cap was bleached (***Video 3***). Cortical recovery in the cap was observed within minutes (***Figure 2—figure supplement 2***), similar to the recovery seen in photobleaching experiments of homogenous cortices (***Figure 1—figure supplement 2***). Despite this rapid turnover at the molecular level, the position of the actin caps remained stable over much longer time scales, typically exhibiting persistent polarization for hours (***Figure 2—figure supplement 3***). Notably, the magnitude of the cortical actin flows and the characteristic length and time scales for polarization observed in our artificial cortices are similar to those seen in developing embryos (***Munro et al., 2004***; ***Cowan and Hyman, 2007***; ***Munro and Bowerman, 2009***; ***Mayer et al., 2010***).

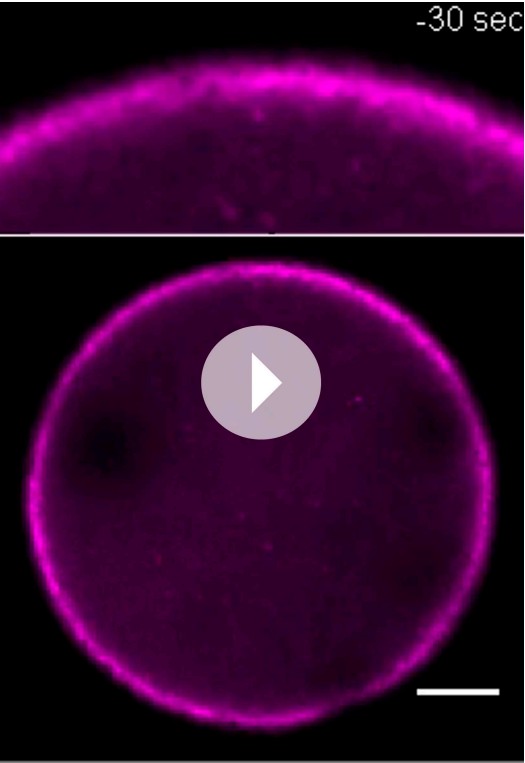

**Video 1**. Cortical recovery after photobleaching. This video shows scanning confocal images (at a single z-plane) of rhodamine-labeled actin within a water-in-oil emulsion incubated at 30°C. A small region of the cortex was photobleached using a high intensity laser pulse. The video follows the dynamics of cortex recovery over several minutes. The lower panel depicts the whole droplet. The field of view is 132 µm wide. The upper panel shows a zoomed area in the proximity of the bleached region. The time relative to the bleaching event is indicated.

To examine whether the temperature dependent contractile behavior we observe in the emulsions is related to the confined geometry as suggested in *Pinot et al. (2012)*, or reflects changes in bulk contractility (*Bendix et al., 2008*; *Kohler et al., 2012*), we designed a bulk assay to assess the behavior of our system. We introduced the same actin machinery into chambers (~100 µm × 100 µm × 5 mm), replacing the amphiphilic ActA with fixed *L. monocytogenes* which express ActA on their surface ('Materials and methods'). At high temperatures, the fixed bacteria remained distributed throughout the sample, exhibiting comet tail motility, and no overall contraction was observed (*Figure 2F*, right). At lower temperatures, we observed rapid contraction of the entire sample, which concentrated actin filaments in the extract and bacterial comet tails into a narrow strip of actin gel (*Figure 2F*, left). The transition between non-contractile and contractile behavior occurred abruptly at ~25°C.

We found that temperature controls the onset of symmetry breaking in our reconstituted actin cortices (*Figure 2A,B*). To investigate whether a temperature shift could lead to symmetry breaking in pre-assembled actin cortices, we first incubated emulsions for 30 min at 30°C to prepare homogenous cortices, and then shifted the temperature to 20°C. Cortical polarity developed as a function of time following the temperature shift (*Figure 2G*, *Figure 2—figure supplement 4*). Initially, the actin network density along the droplet interface was uniform. Within minutes, the homogenous cortical network began to rupture, often in more than one place (*Figure 2—figure supplement 4A*; 22'), leading to the formation of gaps in the actin cortex. After ~30 min, a polar distribution of actin was observed in all emulsions, characterized in most cases by a single polar cap, similar to the distributions measured in emulsions prepared at 20°C (*Figure 2G*). At longer times, we often observed accumulation of actin in the interior of the droplet (near the cap) driven by the actin flows initiated at the interface (*Figure 2—figure supplement 4A*; 75'). The cortical flows also generated a bias in the distribution of the nucleation-promoting factor ActA towards the cap (*Figure 2—figure supplement 4C*). Similar recruitment of nucleation-promoting factors by polymerizing actin networks was previously observed in comet tail motility on soft interfaces (*Boukellal et al., 2004*).

We further characterized the temperature–dependent behavior of the artificial actin cortices as a function of ActA concentration (*Figure 2—figure supplement 5*). As expected, the ActA density at the water–oil interface increased as a function of the initial bulk ActA concentration used (*Figure 2—figure supplement 5A*). The ActA densities at the interface were also higher at 20°C, which can be understood from simple thermodynamic considerations; the dynamic localization of ActA at the interface (*Figure 2—figure supplement 6*) depends on the interplay between enthalpy and entropy which favors localization at lower temperatures. However, within the range of ActA concentrations examined, the intensity of the actin cortices and their symmetry breaking properties were essentially unaffected by the ActA concentration (*Figure 2—figure supplement 5B,C*), suggesting that ActA is in excess. These results suggest that the temperature-dependent symmetry breaking observed reflects the changing properties of the cortical actin networks, rather than differences in ActA localization. The intensity of the actin cortices was found to be higher at lower temperatures, most likely due to the

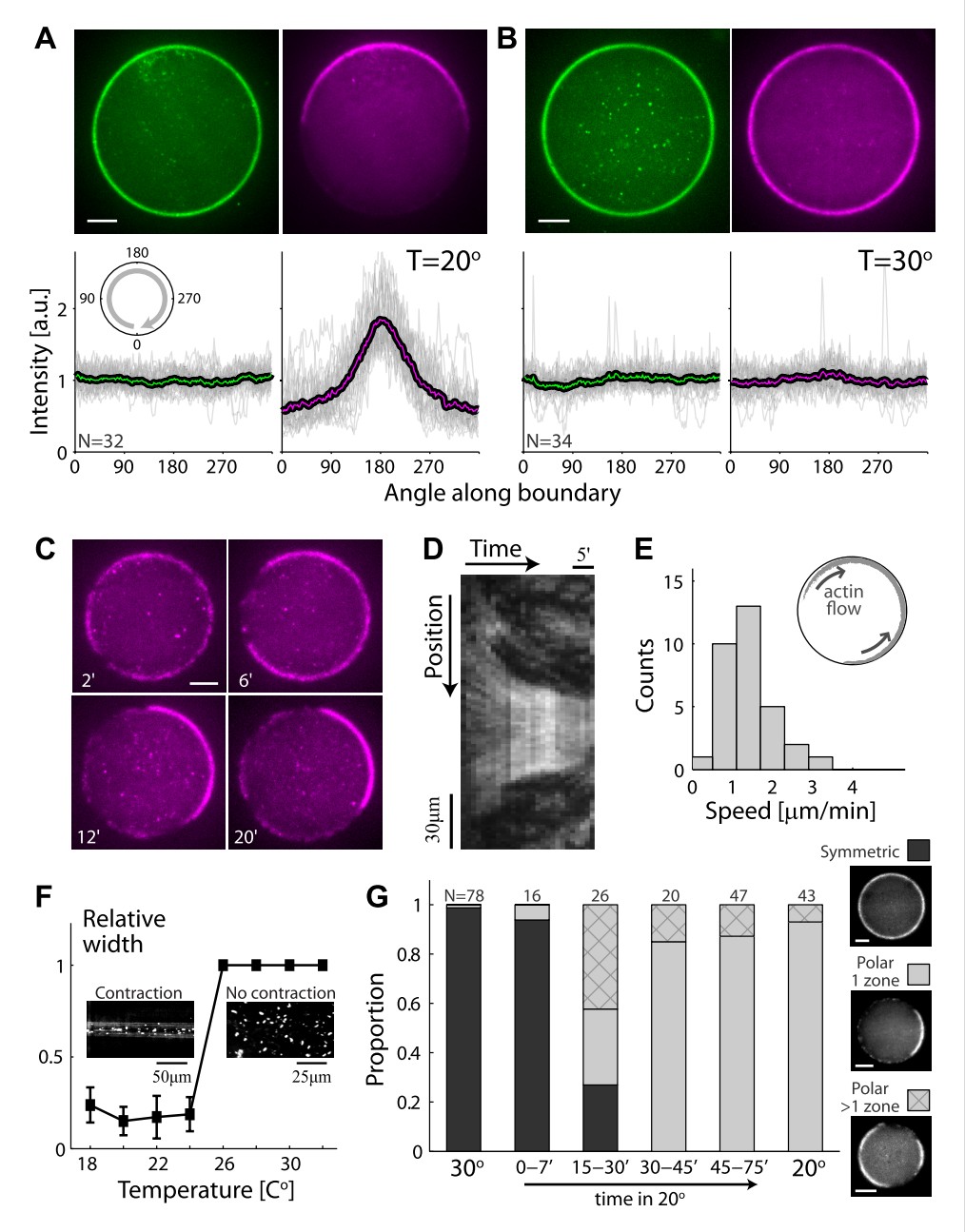

**Figure 2**. Temperature-dependent symmetry breaking in reconstituted actin cortices. (**A** and **B**) Spinning disk confocal images of bodipy-conjugated ActA (top left) and rhodamine-labeled actin (top right) in water-in-oil emulsions incubated at 20°C (**A**) or at 30°C (**B**). A polar actin cortex with a single cap is observed at 20°C, in contrast to the uniform spherical cortex observed at 30°C. The fluorescence intensity profiles as a function of the angle along the contour are shown for different water-in-oil emulsions at 20°C (**A**; bottom panels) or at 30°C (**B**). Data for individual emulsions normalized to the mean intensity in each contour (grey lines) is shown together with the population average (thick line). The actin distributions (bottom right) are peaked at the center of the cap for all the emulsions at 20°C, in contrast to the uniform distributions at 30°C. The ActA distributions (bottom left) are essentially flat at both temperatures. The residual ActA observed outside the polar caps at 30°C suggests that ActA is in excess in our system (**Figure 2—figure supplement 5**). (**C**) Spinning disk confocal images from a time-lapse video (**Video 2**) showing the development of a polar actin cap in an emulsion incubated at 20°C. The time after droplet formation is indicated. Cortex assembly starts uniformly along the interface, and becomes polar within minutes. (**D**) A kymograph of the video in (**C**), showing the fluorescence intensity along the contour of the emulsion
*Figure 2. Continued on next page*

*Figure 2. Continued*

(vertical axis) as a function of time (horizontal axis). The cortical actin flow which slows down as the cap contracts is evident in the kymograph. (**E**) A histogram of the initial cortical flow speeds measured for a population (N = 32) of emulsions followed by time-lapse microscopy during the symmetry breaking process as in (**C**). Inset- a schematic illustration of the cortical actin flow during the symmetry breaking process. (**F**) Bulk assay for actin network contractility. The relative width of the actin containing strip in comparison to the width of the incubation chamber after 30 min is plotted as a function of temperature. At temperatures <25°C the network contracts into a thin strip (left inset), while at higher temperatures (>25°C) no bulk contractility is observed (right inset). (**G**) Samples were incubated for 30 min at 30°C to generate homogenous cortices, and then moved to 20°C. Cortex polarity developed over time, with symmetry breaking often initiating in more than one position (***Figure 2—figure supplement 4A***). The relative proportion of symmetric (dark) and polar cortices (light) with one (bare) or multiple (stripes) actin caps, at different time windows following the temperature shift are shown, in comparison to control samples incubated at 30°C or at 20°C.

The following figure supplements are available for figure 2:

**Figure supplement 1**. Actin and ActA distributions in water-in-oil emulsions at 20°C.

**Figure supplement 2**. Analysis of photobleaching experiments at 20°C.

**Figure supplement 3**. The actin cap remains stable following symmetry breaking.

**Figure supplement 4**. A temperature shift from 30°C to 20°C leads to symmetry breaking.

**Figure supplement 5**. The temperature–dependent behavior of artificial cortices as a function of ActA concentration.

**Figure supplement 6**. ActA dynamics at the interface.

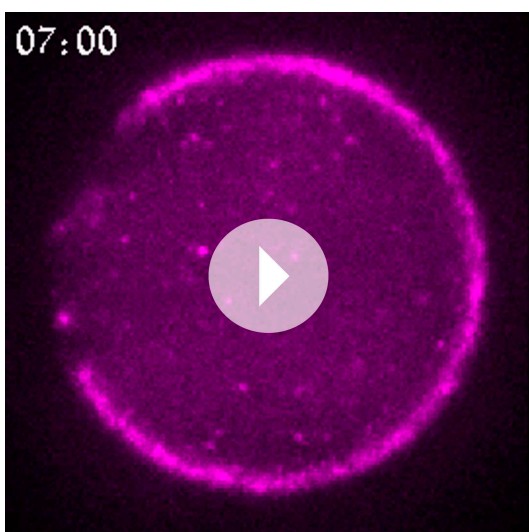

**Video 2**. Actin cortical flow during symmetry breaking. This video shows spinning disc confocal images (at a single z-plane) of rhodamine-labeled actin within a water-in-oil emulsion incubated at 20°C. The actin signal reflects the distribution of actin monomers and filaments. The actin cortex initiates in patches throughout the droplet's interface. Cortical actin flows appear at the interface within minutes, leading to the formation of a polar actin cap. The field of view is 55 µm wide, and the time from droplet formation is indicated.

temperature-induced changes in the balance between actin assembly and disassembly dynamics in our system, also exemplified by the increased lengths of actin comet tails at 20°C compared to 30°C in the same motility mix (data not shown).

## Cortical symmetry breaking is myosin-dependent and requires sufficient network connectivity

Our reconstituted system allowed us to investigate the biochemical requirements for symmetry breaking in the artificial cortices (***Figure 3***). Myosin motors are involved in contractile behavior and symmetry breaking in the cortex of cells (***Munro et al., 2004***; ***Cowan and Hyman, 2007***; ***Munro and Bowerman, 2009***; ***Van Der Gucht and Sykes, 2009***). To examine the role of myosin in our reconstituted system, we depleted myosin II from the *Xenopus* egg extracts by immunodepletion (***Bendix et al., 2008***) ('Materials and methods'). Emulsions made from myosin-depleted extracts did not exhibit cortical actin flows and maintained a symmetric actin shell when incubated at 20°C (***Figure 3A,C***), in contrast to the behavior observed at the same temperature in the presence of myosin. Symmetry breaking was not affected in control experiments with mock immunodepletion (***Figure 3B***). To further establish

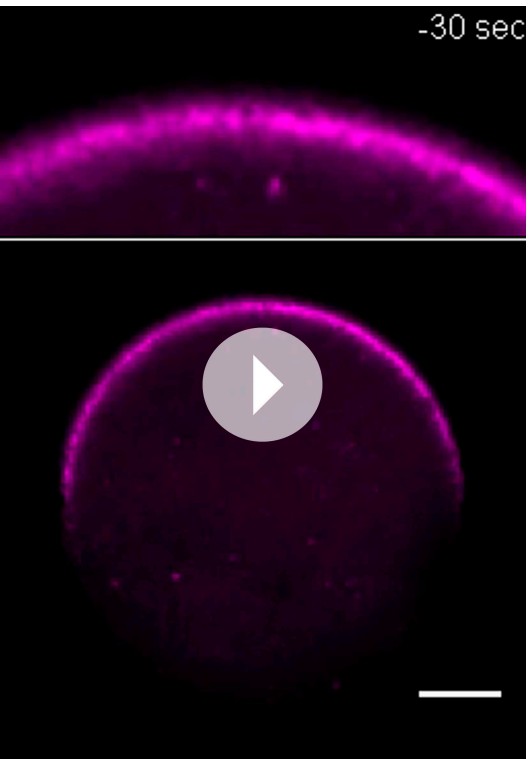

**Video 3.** Cortical recovery after photobleaching in an asymmetric actin cap. This video shows scanning confocal images (at a single z-plane) of rhodamine-labeled actin within a water-in-oil emulsion incubated at 20°C. A small region at the tip of the actin cap was photobleached using a high intensity laser pulse. The video follows the dynamics of cortex recovery over several minutes. The lower panel depicts the whole droplet. The field of view is 132 µm wide. The upper panel shows a zoomed area in the proximity of the bleached region. The time relative to the bleaching event is indicated.

the essential role of myosin in the symmetry breaking process, we added different amounts of purified myosin to myosin–depleted extracts and showed that the symmetry breaking is restored (*Figure 3B,C*). The cap morphology was dependent on the amount of added myosin; as the concentration of myosin increased, we observed more cortices with multiple caps, rather than a single cap (*Figure 3C*). Moreover, at even higher myosin concentrations (0.66 µM) most of the cortices appear fractured in many places with multiple small interconnected cortical patches which exhibited jittery motion. The decrease in the typical size of cortical patches with increasing amounts of myosin is probably due to the decrease in the contractile unit size as a function of myosin concentration (*Thoresen et al., 2013*). Overall our results suggest that the symmetry breaking observed in our artificial cortices is driven by the same myosin-induced mechanism found in cells (*Munro et al., 2004*; *Mayer et al., 2010*).

The contractile behavior of actin networks is also dependent on the degree of crosslinking within the network; Sufficient network connectivity is required to enable myosin motors to generate forces rather than merely induce sliding of filaments, whereas excessive crosslinking stiffens the network and prevents contraction (*Bendix et al., 2008*; *Kohler et al., 2012*). We used two different kinds of actin crosslinkers which are known to localize to cortical actin networks in vivo (*Charras et al., 2006*), α-actinin which preferentially generates anti-parallel actin bundles and filamin which is a more flexible crosslinker that can bind to disordered actin networks. We showed that symmetry breaking can be induced at 30°C by adding crosslinkers (*Figure 3D,E*). Increasing α-actinin concentration led to the formation of polar cortices in a concentration dependent manner. At low concentrations (<4 µM) the cortices remained largely symmetric, at intermediate concentrations (4–6 µM) we mostly observed the formation of a single actin cap, while at higher concentrations (8 µM) we often observed multiple actin domains. Addition of filamin (4 µM) also led to the formation of a polar actin cap. Despite the different properties of these crosslinkers, they were both successful at inducing symmetry breaking, highlighting the importance of the overall connectivity of the network rather than the detailed characteristics of individual crosslinkers.

## Cell-like phenomena in the artificial cortices

Interestingly, our artificial actin cortices exhibit additional cell-like behaviors (*Figure 4*). The cortex, which is under contractile stress, could spontaneously detach from the interface locally (*Figure 4A*; *Video 4*), as observed in vivo during cellular blebbing (*Charras et al., 2006*). Detachment of the cortex did not lead to outward bulging of the interface due to the high surface tension of the water–oil interface in our system (~700 pN/µm, 'Materials and methods'). As in cells (*Charras et al., 2006*), the actin cortex healed over time and a new cortex assembled at the interface within minutes. In addition, the asymmetric actin cortices generated polar forces which could induce reorganization in the interior of the droplet and global deformation of the droplet's shape (*Figure 4B,C*; *Video 4*). Such deformations were observed in ~10% of the emulsions, and their incidence increased after longer incubation

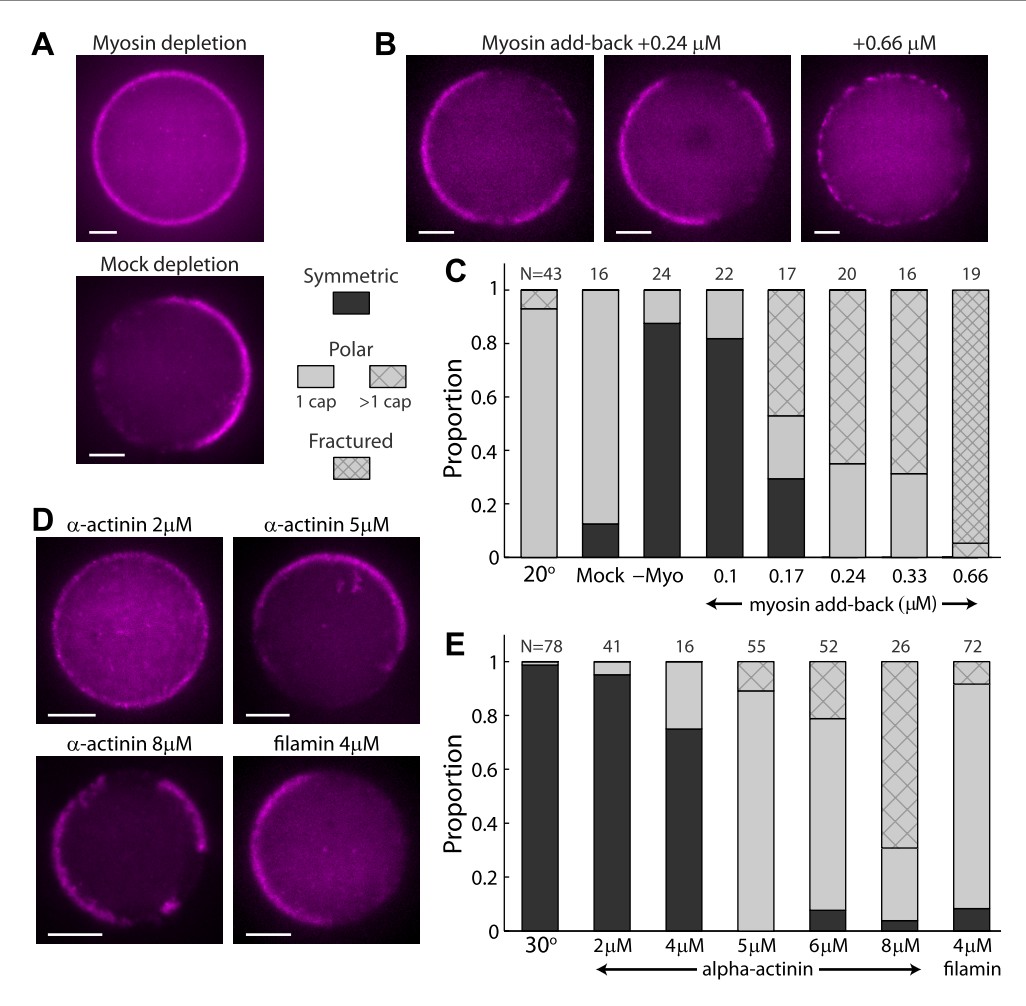

**Figure 3**. The effect of myosin and crosslinkers on symmetry breaking. (**A**) Spinning disk confocal images of actin cortices in water-in-oil emulsions made with myosin–depleted (upper panel) or mock-depleted (lower panel) extracts and incubated at 20°C. Myosin depletion eliminates symmetry breaking, whereas a mock depletion does not. (**B**) Spinning disk confocal images of actin cortices made with myosin-depleted extracts supplemented with different amounts of purified myosin. At intermediate myosin concentrations (0.24 µM; left images) the actin cortices typically have one or few actin caps. At higher myosin concentrations (0.66 µM; right image) the actin cortices appear fractured with many discrete puncta along the interface. (**C**) Bar plot showing the relative proportion of symmetric (dark), polar cortices (light) with one (bare) or multiple (stripes) actin caps, or fractured cortices (dense stripes). Data is shown for a control sample of emulsions at 20°C, in comparison to samples prepared at the same temperature with myosin-depleted, mock-depleted extracts or myosin-depleted extracts supplemented with different amounts of purified myosin. (**D**) Spinning disk confocal images of actin cortices formed with extracts supplemented with different amounts of α-actinin or filamin crosslinkers and incubated at 30°C. (**E**) Bar plot showing the relative proportion of symmetric and polar cortices (as in (**C**)) for samples of emulsions made with different amounts of crosslinkers (2–8 µM α-actinin; 4 µM filamin) relative to a control sample at 30°C. The addition of crosslinkers induces symmetry breaking at 30°C. Note the large fractions of emulsions at higher α-actinin concentrations that exhibit multiple actin caps. Scale bars: 10 µm.

times compared to the initial polarization event. Similar reorganization and changes in global morphology have been observed in cells and embryos (**Lecuit and Lenne, 2007**; **Munro and Bowerman, 2009**). The shape deformation of the droplet can be used to measure the forces generated by the cortical actin network (**Boukellal et al., 2004**; **Figure 4C**). The high curvature at the tip of the actin cap indicates that the cytoskeleton exerts a net protrusive force there, which we estimate to be ~20 pN/µm$^2$ (**Figure 4C**). Away from the tip, on the sides of the actin cap, the local curvature is lower implying that

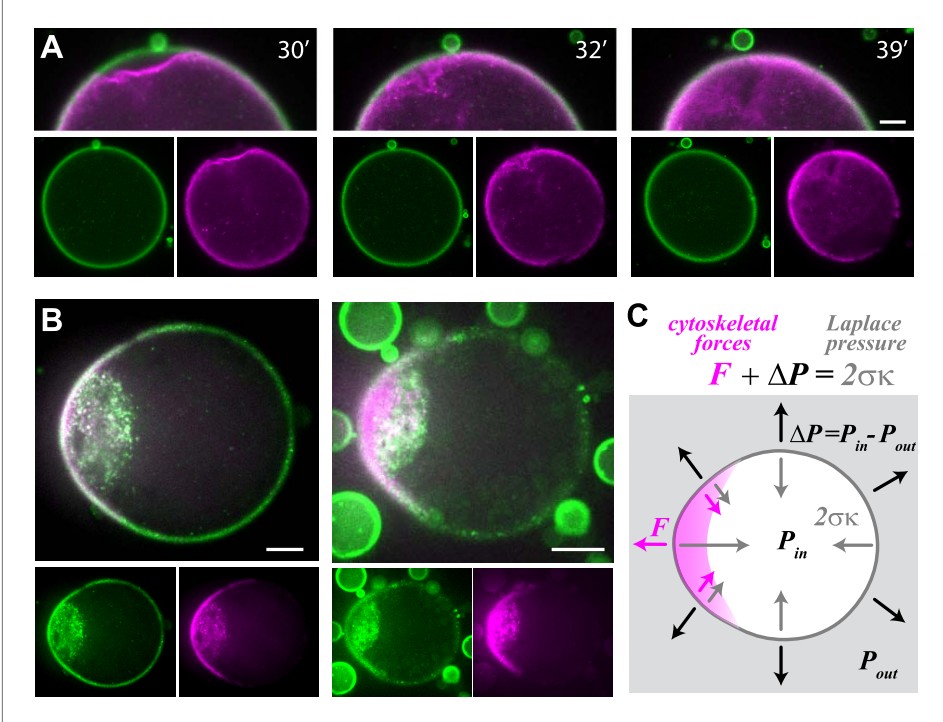

**Figure 4**. Cell-like behavior of reconstituted actin cortices. (**A**) Scanning confocal images from a time-lapse video (**Video 4**) showing local detachment of an actin cortex from the interface followed by regrowth of a new cortex within minutes. The time after droplet formation is indicated. Images of the bodipy-conjugated ActA (green) and rhodamine-labeled actin (magenta) are depicted, together with an overlay of the detachment region (top). (**B**) Spinning disk confocal images of two representative water-in-oil emulsions incubated at 20°C, exhibiting polar actin cortices and deformed morphology. Images of the bodipy-conjugated ActA (green), rhodamine-labeled actin (magenta) and their overlay are shown. The asymmetric actin cortices generate polar forces which induce deformation of the droplets. Note the formation of a cytoplasmic concentration of actin and ActA near the deformed cap. The ActA distribution is polar in the example shown on the right, where the overall amount of ActA appears lower (**Figure 4—figure supplement 1**). (**C**) A schematic illustration of a deformed droplet and the forces at its interface. The cytoskeletal forces $F$ and the pressure difference $\Delta P$ at the interface are balanced by the Laplace pressure which is proportional to the surface tension $\sigma$ and the local mean curvature $\kappa$, so that $F + \Delta P = 2\sigma\kappa$. The pressure difference can be estimated from the curvature outside the cap and the known surface tension, since the cytoskeletal forces are assumed to be zero outside the cap. The protrusive force within the cap was estimated from the difference in the local curvatures in the cap region and outside the cap: $F = 2\sigma\kappa_{cap} - \Delta P = 2\sigma\left(\kappa_{cap} - \kappa_{no\,cap}\right)$. Scale bars: 10 µm.

The following figure supplements are available for figure 4:

**Figure supplement 1**. Actin and ActA distributions in a deformed droplet.

the cytoskeleton is exerting pulling forces on the interface. These pulling forces are likely responsible for the accumulation of actin and ActA near the cap (**Figure 4B**, **Figure 4—figure supplement 1**).

## Discussion

In this work we present a novel reconstituted system that self-organizes into dynamic actin cortices which exhibit spontaneous symmetry breaking in the absence of any directional cue. Our system integrates actin assembly at the interface, actin turnover, myosin motor activity and crosslinkers within a confined geometry, and displays interesting cell-like behavior. The water-in-oil emulsions provide an easy and reproducible approach to generate cell-like compartments (**Tawfik and Griffiths, 1998**; **Pinot et al., 2012**; **Good et al., 2013**; **Hazel et al., 2013**). Giant vesicles (GUVs) have also been used as compartments for reconstituting cellular phenomena (**Noireaux and Libchaber, 2004**; **Pontani et al., 2009**; **Carvalho et al., 2013b**). Both approaches have their advantages and limitations;

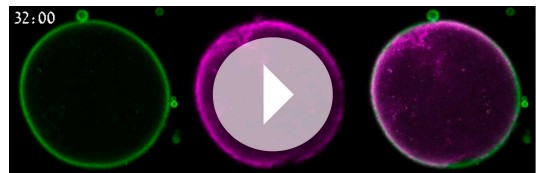

**Video 4**. Cytoskeletal forces in the actin cortex lead to bleb-like cortex detachment and shape deformation. This video shows scanning confocal images (at a single z-plane) of bodipy-conjugated ActA (left), rhodamine-labeled actin (center) and an overlay (right), within a large water-in-oil emulsion incubated at 20°C. The contractile forces within the cortex led to detachment of the cortex from the interface. The cortex recovered within minutes by regrowth of a new cortex at the interface. The polar forces generated by the cortex also led to shape deformation which developed over time. The field of view is 109 μm wide, and the time from droplet formation is indicated.

encapsulation in GUVs is more challenging and the GUVs limited stability in cell extracts can be problematic, while the use of oil-based emulsions limits the applicability of various cell permeable biochemical drugs (e.g., blebbistatin) which tend to be hydrophobic and segregate into the oil phase. In addition, the actin machinery within emulsions appears more sensitive to phototoxicity effects, probably due to elevated oxygen levels within the emulsions compared to bulk. Despite these limitations, our system offers new possibilities for studying cytoskeletal dynamics; we are able to vary the identity and concentrations of actin nucleators, myosin motors and crosslinkers within a controlled environment, and follow their dynamics over time. Using this novel reconstituted system we show that the assembly of dynamic actin cortices requires localization of nucleation factors at the interface, and identify the conditions for cortical symmetry breaking. Specifically, we show that symmetry breaking requires myosin motors and sufficient network connectivity, but does not depend upon pre-patterned localization of actin nucleators, the involvement of microtubules (*Munro et al., 2004*), or any local changes in the properties of the interface.

The dynamics of the artificial cortices are temperature–dependent; a shift by a few degrees leads to a large qualitative change in the behavior of the system, going from homogenous cortices at high temperature to asymmetric actin caps at low temperature (*Figure 2*). Obviously, temperature is a convenient control parameter. Experimentally, this allowed us to show that cortical polarization can be induced at high temperature by adding crosslinkers and prevented at low temperature by depleting myosin (*Figure 3*), implying that symmetry breaking in the artificial cortices is driven by contraction of the cortical actomyosin network. This contractile behavior observed at low temperatures is consistent with previous experiments in cell extracts which were all done at temperatures below 20°C (*Bendix et al., 2008*; *Field et al., 2011*; *Pinot et al., 2012*). At 30°C the cortical network appears weakly crosslinked, and hence unable to transmit the contractile stresses generated by myosin motor activity. Visualization of the ultrastructure of the actin networks could provide additional insight into the mechanism of symmetry breaking. However, analysis of actin structures by electron microscopy within cell-like compartments is technically challenging due to difficulties in fixation and sample preparation. We hypothesize that the transition to contractile behavior at lower temperature is primarily due to an increase in the intrinsic crosslinking activity in the system. This increase could emerge from longer dwell times of individual myosin motor heads at low temperature, which would turn the multi-motor myosin filaments into more effective crosslinkers. Alternatively, decreasing temperature could increase the affinity of endogenous actin crosslinkers. Our observations are analogous to recent results, which revealed a sharp transition in the contractile behavior of active actin networks as a function of pH (*Kohler et al., 2012*). Overall these findings indicate that actin cortices are close to an instability threshold; small changes in the composition of the system, or in the activity of its components, can lead to dramatic changes in the global characteristics of the system. We believe that this reflects a general organizational principle in cellular systems, whereby cells are often posed near an instability threshold to enable rapid responses to changing conditions.

As individual modules within the cells are being unveiled at the molecular level, understanding their integration at the whole-cell level is becoming a central challenge in cell biology. Reconstituting different functional modules together in a meaningful way can be invaluable in that respect, but such reconstitutions have proven to be challenging experimentally. Here we report the successful integration of different cytoskeletal modules into a single reconstituted system which recapitulates several cellular phenomena including cortical symmetry breaking (*Figures 2 and 3*), polar force generation, and blebbing (*Figure 4*). Our reconstituted system provides novel insights into the complex behavior of the actin cortex in living cells, presenting new opportunities for investigating cortical dynamics,

both experimentally and theoretically (*Joanny et al., 2013*). More generally, our system opens the way for future exploration of cytoskeletal dynamics within cell-like compartments, and the integration of additional modules, such as protein expression systems (*Noireaux and Libchaber, 2004*) and cell cycle (*Good et al., 2013*; *Hazel et al., 2013*), towards the realization of artificial cells that can move and divide. Advances in this direction will promote our understanding of basic cellular functions, as well as present ample opportunities for future application in therapeutics and bioengineering.

## Materials and methods

### Proteins and reagents

Actin was purified from chicken skeletal muscle and labeled in filamentous form on either amines or cysteine 374 with tetramethylrhodamine iodoacetamide (#T6006; Molecular Probes, Grand Island, NY), Tetramethylrhodamine Succinimidyl Ester (#T6105) or AleaFluor647 Succinimidyl Ester (#A20006) at a ratio of 1:10 using standard protocols. The behavior of the actin cortices was similar with the different types of labeled actin used (*Figure 1—figure supplement 3*). The purified actin was stored in G-buffer (10 mM Tris–HCl pH 8.6, 0.1 mM DTT, 0.2 mM ATP, 0.1 mM CaCl2) on ice for up to 3 weeks, since the assembly of the cortices was sensitive to the quality of the actin.

Chicken skeletal myosin was purified according to standard protocols and stored lyophilized in high salt buffer (500 mM KCl, 50 mM Hepes pH 7.5, 5% sucrose). Before use, dead myosin heads were removed by spinning the myosin hexamers in the presence of 1 mM ATP and preformed actin filaments, as described in *Thoresen et al. (2011)* but without adding phalloidin.

ActA-His-Cys was purified from strain DP-L4363 of *L. monocytogenes* (a gift from Julie Theriot, Stanford University) expressing a truncated actA gene encoding amino acids 1–613 with a COOH-terminal six-histidine tag replacing the transmembrane domain and containing an additional cysteine amino acid (*Welch et al., 1998*). ActA was conjugated through a heterobifunctional linker LC-SMCC (#22362; Thermo Scientific/Pierce, Rockford, IL), to poly-D-lysine (#P0296; Sigma–Aldrich, St. Louis, MO) labeled with ~6–8 molecules of Bodipy FL-X-SE (#D6102; Molecular Probes) per peptide. The poly-D-lysine was first incubated with the linker and Bodipy at room temperature for 1 hr. The reaction was quenched with 1M Tris pH 7.5. Reduced ActA-His-Cys was added to the mixture and incubated for 2 hr at room temperature. The unbound reagents were removed by dialysis against XB buffer (100 mM KCl, 0.1 mM CaCl2, 2 mM MgCl2, 5 mM EGTA, 10 mM K-HEPES pH 7.7).

α-actinin was purchased from Cytoskeleton Inc. (Denver, CO), and reconstituted to final concentration of 40 µM with water. Filamin was purchased from Prospec (East Brunswick, NJ), dialyzed against Hepes pH 7.5 and reconstituted to final concentration of 20 µM in XB buffer with 50 mM Hepes.

### Cytoplasmic extracts preparation and manipulation

Concentrated M-phase extracts were prepared from freshly laid *Xenopus laevis* eggs as previously described (*Cameron et al., 1999*). Immunodepletion of myosin II was performed as in *Bendix et al. (2008)* with minor modifications. In brief, anti-myosin antibody raised against a C-terminal peptide of *Xenopus* myosin II-A heavy chain (a gift from Aaron Straight, Stanford University) or random rabbit IgG (#011-00-003; Jackson Laboratories, Bar Harbor, ME) were covalently bound to protein-A Dynabeads (#100-02; Dynal Biotech, Grand Island, NY) and resuspended in XB. The extract was incubated with the conjugated Dynabeads for two sequential depletion rounds of 10 min each at RT, and used immediately.

### Emulsion preparation

A motility mix was prepared by mixing the following: crude extract (4 µl), XB containing 30 mM MgCl$_2$ (4 µl), 0.13–0.2 mM tetramethylrhodamine labeled actin in G-buffer (1 µl), 20 × ATP regenerating mix (150 mM creatine phosphate, 20 mM ATP, 20 mM MgCl2 and 20 mM EGTA) and 30 µM bodipy-conjugated ActA (0.5 µl each). We estimate the total actin concentration in the motility mix to be ~25 µM. Emulsions were made by adding 1% (vol/vol) motility mix to mineral oil (Sigma) containing 4% Cetyl PEG/PPG-10/1 Dimethiocone (Abil EM90; Evnok Industries, Essen, Germany) and stirring for 2 min at room temperature. The oil and surfactant mixture was degassed under vacuum overnight prior to use to reduce phototoxicity. Samples were made in chambers assembled from a glass slide and a coverslip which was passivated with trichloromethyl silane (Sigma), and sealed with vaseline:lanolin:paraffin (at 1:1:1). Samples were either imaged immediately (for time-lapse videos) or incubated for 30 min at the indicated temperature and then imaged for population analysis. The surface tension between the aqueous and the oil phase was measured using the pendant-drop assay. An extract containing droplet

was injected into an oil cuvette, and the droplet was imaged from the side to determine its shape. The surface tension was determined by analyzing the shape of the droplet using standard procedures.

## Microscopy and analysis

Emulsions were imaged on a 3I spinning disk confocal microscope running Slidebook software, or a laser scanning confocal microscope (Zeiss LSM 700) running ZEN 2009 software, using a 63 × oil objective (NA = 1.4). Images were acquired using 488 nm and 561 nm laser illumination and appropriate emission filters in a temperature controlled incubator. Images on the spinning disk confocal were collected with an EM-CCD (QuantEM; Photometrics, Tucson, AZ). Images of emulsions were typically taken at a single plane at ~1/3 of the emulsion height, since imaging at the mid-plane or higher suffered from optical artifacts due to the differences in the indices of refraction between the water and oil phase. Bright field images were taken at the mid-plane. Quantitative image analysis was done using the Celltool package developed by Zach Pincus (*Pincus and Theriot, 2007*) and custom code written in Matlab. Analysis was done on emulsions with a diameter of 30–80 µm. Actin contractility in bulk was imaged using either a 20 × air objective (NA = 0.6) or a 40 × oil objective (NA = 1.3).

FRAP experiments were done on a laser scanning confocal microscope (Zeiss LSM 700) using a 10 mW 555 nm laser. Bleaching was done on a ~1.5 × 1.5 µm region of a cortex either at the bottom of an emulsion or at its side. FRAP analysis was done using the ZEN software by fitting the recovery to a single exponent. An unbleached cortical region was used as a reference region in the analysis.

## Acknowledgements

We are grateful to Anne Bernheim, Yariv Kafri, Erez Braun and Shlomit Yehudai- Resheff for many helpful discussions. We thank Amnon Harel and Boris Fichtman for providing *Xenopus* eggs and useful advice. We thank Galia Blum for suggesting to use Bodipy, Denis Cottinet and Jérôme Bibette for advice on surfactants, Julie Theriot for ActA constructs, and Aaron Straight for myosin antibodies. We thank Maya Malik-Garbi for the surface tension measurements of the water–oil interface, and Gidi Ben Yoseph for superb technical help and support. We thank Christine Field for advice on extract preparation, and Michael Murray for advice on myosin purification. We thank Alex Mogilner, Arbel Artzy-Shnirmann and all the members of our lab for comments on the manuscript. This research was supported by a Starting Independent Researcher Grant and an International Reintegration Grant from the European Research Council to KK.

## Additional information

### Funding

| Funder | Grant reference number | Author |
| --- | --- | --- |
| European Research Council- starting grant | 203077 | Kinneret Keren |
| European Research Council- reintegration grant | 239229 | Kinneret Keren |

The funder had no role in study design, data collection and interpretation, or the decision to submit the work for publication.

### Author contributions

EAS, KK, Conception and design, Acquisition of data, Analysis and interpretation of data, Drafting or revising the article

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
