## [Decision Letter]

[Editors’ note: this article was originally rejected after discussions between the reviewers, but the authors were invited to resubmit after an appeal against the decision.]

Thank you for choosing to send your work entitled “Symmetry breaking in reconstituted actin cortices” for consideration at *eLife*. Your full submission has been evaluated by a Senior editor and 3 peer reviewers, one of whom is a member of our Board of Reviewing Editors, and the decision was reached after discussions between the reviewers. We regret to inform you that your work will not be considered further for publication.

The specific comments/concerns of the 3 expert referees are enclosed. As you will see, all three of them believe the system you have established is very interesting and can be a powerful tool with which to study symmetry breaking. However, they all concur that there are significant technical concerns and the work is not advanced enough to provide a mechanism for symmetry breaking with sufficient depth for publication at this stage in *eLife*.

*Reviewer*
*1:*

The manuscript by Shah and Keren addresses the important topic of symmetry breaking, a key to wide range of biological processes. They establish a water-in-oil emulsion made with a fluorescent version of the Listeria ActA, Xenopus egg extract and fluorescent actin and first show that actin polymerization and an actin cortex develops around at the periphery of the interface between the oil-aqueous media. Interestingly, they find that at lower temperatures (approx. 20°C) the actin network breaks symmetry and generates a cap, which does not happen at higher temperatures. Incubation with myosin II antibodies blocks symmetry breaking and crosslinking is required. Although the establishment of this new reconstituted cortex is interesting and the result with myosin II potentially the novel finding the work is not developed enough to warrant publication in *eLife*. In particular, several conclusions are made too loosely. Also, while the water-in-oil approach is new, it does not go well over and above the system characterized by Theriot and van Oudenaarden groups on symmetry breaking in ActA-coated beads as far as an in vitro non-cell system is concerned.

Some examples of issues with the work:

1) It will be nice to have a proper characterization of what else is in the reconstituted actin cortex at 20°C and 30°C. Such an analysis will provide an explanation for why addition of crosslinkers promotes symmetry breaking at higher temperatures. I am not sure how technically feasible this would be to preserve these structures for further manipulation. But such information is essential before strong conclusions, as in the manuscript, can be made.

2) Likewise, it will be important to know if the actin cytoskeleton is composed of branched actin or actin cables. I recognize Arp2/3 generates branched actin, but through cofilin function they can be modified into linear cables. Is there a way to process the artificial cortices for EM to get at this question?

3) If these are indeed branched actin, some discussion is required on the mechanism of myosin II based symmetry breaking of branched actin networks.

4) The myosin II immunodepletion experiment is also cursory. No western blots are shown to know how much of myosin II is depleted. Also, it is possible that other factors in the myosin II immune complex may be responsible for the effect. Perhaps a simple experiment with blebbistatin can help. I am not sure if blebbistatin works on Xenopus myosin II. If not, they can first supplement the extract with human myosin II (after depletion of Xenopus myosin II) followed by blebbistatin addition. Such an experiment of humanized-Xenopus extract can answer all concerns mentioned in this point.

*Reviewer*
*2:*

This manuscript by Shah and Keren presents a new in vitro reconstituted system to study the actomyosin cortex. In this work, the authors demonstrate that the presence of Xenopus cytoplasmic extract and the actin promoting factor ActA from Listeria in water-in-oil droplets lead to the assembly of an artificial, dynamic actin cortex. Importantly, these artificial cortices undergo spontaneous symmetry breaking, which shows that this reconstituted system could be used to further study and understand the mechanisms underlying symmetry breaking. This paper is generally well structured and well written and provides a nice account of this potentially very useful minimal system for studying the actomyosin cortex.

The investigators provide some limited insight into the symmetry breaking mechanism, with data suggesting that myosin-II and actin crosslinking are important for this process. However, this report serves mainly to describe the novel and interesting reconstituted system.

Major points:

1) A key conclusion of the paper is that myosin contractility can drive symmetry breaking in an actin cortex reconstituted in a droplet. This is a very interesting finding. However, the conclusion relies on a single experiment: the immunodepletion of myosin II-A heavy chain in the extracts. It is unclear how much myosin is really depleted using this method, other myosins are likely to be present in the extract, and there is no experimental evidence that myosin really localizes to the cortex in this system. To truly conclude on the importance of contractility, it seems essential to use some alternative method to test the functional importance of myosin activity. Is it possible to use drugs (such as blebbistatin) in this system? Alternatively, could the localization and dynamics of myosin be imaged during symmetry breaking?

2) The authors allude several times to a dynamic cortex, stating that actin polymerization and depolymerization are important for symmetry breaking. However, this is not investigated directly. FRAP experiments to quantitatively measure actin dynamics would help to better describe the behavior of this system and to support the authors' conclusions on the importance of actin dynamics.

3) It is puzzling that droplet deformation is observed in only 10% of the polarized droplets. The authors speculate that this might be due to the relatively high tension of the droplet interface, which they estimate at 700 pN/μm. Could they give details on how this tension was measured? This tension is indeed rather high, but not that much higher than tension measured by the same group in migrating keratocytes (Lieber et al. 2013). Was any deformation observed in the droplets at higher temperature, where polarization was induced by addition of crosslinkers? One would expect that interfacial tension would be lower at a higher temperature.

*Reviewer*
*3:*

In this manuscript the authors demonstrate symmetry breaking and polarization in a reconstituted actin cortex. They establish a system where actin polymerization and cortex formation is induced by recruitment of an amphiphilic nucleator (ActA) to the interface of a water-in-oil emulsion – with the water phase represented by M-phase extracts from Xenopus eggs. Interestingly, they then observe spontaneous symmetry breaking and formation of a polar actin cap when emulsions were incubated a lower temperature (20 instead of 30°C). By depletion and addition of proteins to the egg extracts the authors demonstrate that myosin and the right amount of actin crosslinkers are required for cortex polarization.

The introduced biomimetic system using water-in-oil emulsions is very elegant and promising. Cortex formation and symmetry breaking appear to be remarkably similar to previously shown behaviors in cells or whole embryos. The experiments in this study also nicely show the advantages and tools available in the established system.

My main issue with this manuscript is that it does not go any further than the demonstration of an interesting technique. No conceptual or biological advances are presented and it is also not clear to me from the Discussion what the authors hope to achieve with their system in the future (except create artificial cells, which is very long term goal). Therefore, despite the very well written text and the elegance of the new system, I am not convinced that this manuscript has the required scope for *eLife*.

Specific comments:

1) The involvement of myosin and crosslinkers follows concepts that are becoming increasingly established (see for example also “Crosslinking proteins modulate the self-organization of driven systems”, 2013, Schaller, et al. Soft Matter or “Mechanical properties of reconstituted actin networks at an oil-water interface determined by microrheology”, 2012 Ershov et al. Soft Matter).

2) Water-oil emulsions have been previously used as biomimetic system for the study of cell-cell adhesion (Biomimetic emulsions reveal the effect of mechanical forces on cell-cell adhesion, 2012, Pontani et al. PNAS). This study should be cited as precedent for the technique and could be included in the Discussion regarding potential extension for artificial cells.

3) The system should allow comparably simple analysis of cap formation. The authors should therefore provide more information about the exact time course of cortex polarization: parameters include persistence of clusters, rate of movement (not only the initial flow speeds). In particular show formation of initial clusters with better temporal resolution.

4) Use FRAP to characterize lateral mobility and turnover of the cortex (actin and ActA).

5) Labeling and movies of myosin would provide a much better understanding of the observed behaviors

6) In Video 2 it appears like cytosolic patches are moving in coordination with the cortical cap. It is possible that the actin network also extends into the extract and actually represents a 3D meshwork. This would be closer to the previous studies on simple reconstituted actin networks. Can EM images be obtained from fixed slices to test this?

7) What is the nature of internal actin/ActA structures in deformed drops? Does actin cover small lipid droplets? Can a lipid dye be used as counter stain?

8) Easy additional control experiments would be inhibition of Myosin by blebbistatin (to show that myosin activity is needed) and depletion of specific crosslinkers from extracts. Why was only one concentration used for filamin? Show dose-response curves to better compare (maybe crosslinkers are not all equal). Can the published effect of pH on cortex organization (Bausch lab) be reproduced in the new system?

[Editors’ note: what now follows is the decision letter after the authors submitted for further consideration.]

You will be happy to see that the referees have indeed reviewed not only your rebuttal letter, but also the entire manuscript, and are now positive about the manuscript and have made further suggestions. The two major questions revolve around quantification of the FRAP experiments and some rewriting to explore potential uses and limitations of the system you have developed.

The relevant points of the reviewers are enclosed verbatim for your consideration:

*Reviewer*
*1:*

In this revised version of their manuscript the authors have addressed most of my previous concerns. In particular the add-back of myosin and the FRAP analysis provide important support for their conclusions. However, my main concern about a lack of conceptual advance in this study was not addressed by the authors at all – neither by experiments nor in writing – and therefore still remains valid.

Also, they argue that a better quantification and resolution of the cap formation process (which was my main suggestion to possibly achieve a conceptual understanding of the symmetry breaking process) is hindered by phototoxicity. This then seems to be a significant limitation of the new system.

In summary, I would in principle support publication provided that the authors include a paragraph in their Discussion that clearly states limitations of the system (difficult fixation, phototoxicity, problems with lipophilic drugs). In addition, possible approaches to gain new fundamental insights into the symmetry breaking mechanism using the emulsion system should be identified and discussed.

*Reviewer*
*2:*

The authors did a good job at addressing the reviewers' comments and the paper now presents a thorough description of this very interesting experimental system. In particular, the myosin addition experiment indeed demonstrates that myosin drives symmetry breaking in this system. Are the patches formed upon addition of large amounts of myosin (Figure 3, right panel) dynamic, like in the *C. elegans* cortex, or static?

I have some concerns about the photobleaching experiments, which are important for the message of the paper: in the absence of quantification, it is unclear how representative they are. Some quantification (e.g., average recovery time) and an example of recovery curve for the cap and the uniform cortex, should be provided.

Also, did photobleaching never induce symmetry breaking? Photobleaching of actin networks in vitro has been reported previously to induce symmetry breaking (possibly by photo-induced network disruption, e.g., van der Gucht et al. 2006).

---

## [Author Response]

[Editors’ note: the author responses to the first round of peer review follow.]

Reviewer 1:

*While the water-in-oil approach is new, it does not go well over and above the system characterized by Theriot and van Oudenaarden groups on symmetry breaking in ActA-coated beads as far as an* in vitro *non-cell system is concerned*.

We respectfully disagree with this statement. The symmetry breaking of the actin network studied by Theriot and van Oudenaarden as well as by other groups (e.g., Sykes group, Mullins group) is entirely different. While in both cases the system exhibits spontaneous symmetry breaking in vitro, the mechanism is not the same. In particular, the force generating mechanism is different; actin polymerization is essential and sufficient for symmetry breaking of beads and the formation of comet tails. This is demonstrated in reconstitution experiments which contain only actin nucleators and actin polymerization factors, but do not contain myosin (20). The symmetry breaking we observe in artificial cortices, which is similar to the symmetry breaking observed in a developing embryo, is dependent on myosin motors as a force generator. As shown by us in vitro and by others in vivo (27), myosin is essential for this process and actin polymerization alone does not lead to symmetry breaking. As such, the conditions for symmetry breaking and the mechanisms involved are substantially different. Thus, while both systems demonstrate interesting emergent behavior and polarize spontaneously, we do not agree that the systems are equivalent as far as a non-cellular system for symmetry breaking.

*Some examples of*
*issues with the work:*

*1) It will be nice to have a proper characterization of what else is in the reconstituted actin cortex at 20°C and 30°C. Such an analysis will provide an explanation for why addition of crosslinkers promotes symmetry breaking at higher temperatures. I am not sure how technically feasible this would be to preserve these structures for further manipulation. But such information is essential before strong conclusions, as in the manuscript, can be made*.

*2) Likewise, it will be important to know if the actin cytoskeleton is composed of branched actin or actin cables. I recognize Arp2/3 generates branched actin, but through cofilin function they can be modified into linear cables. Is there a way to process the artificial cortices for*
*EM to get at this question?*

*3) If these are indeed branched actin, some discussion is required on the mechanism of myosin II based symmetry breaking of branched actin networks*.

We agree with the reviewer that it would be insightful to characterize the ultrastructure of the artificial cortices in our system, and reveal the detailed composition and structure of the actin cortices. However, the accessibility of the reconstituted cortices inside emulsions to further manipulation and EM analysis is limited. It is not possible to extract the cortices from the emulsions while preserving their structure. Moreover, while it is possible to use cryo-fixation to prepare samples of cortices inside intact emulsions, imaging the cortices in these thick two-phase systems is problematic. Thus, EM analysis of these system using currently available sample preparation techniques is not feasible.

With regards to the ability of myosin II to generate contraction in branched networks. Very recent reports on branched actin networks reconstituted on lipid bilayers (Carvalho, Sykes et al. Philos Trans R Soc Lond B Biol Sci 2013) describe contraction of branched actin networks, so myosin II appears to act both on bundled and on branched networks. We have added citation to this paper in the revised manuscript.

*4) The myosin II immunodepletion experiment is also cursory. No western blots are shown to know how much of myosin II is depleted. Also, it is possible that other factors in the myosin II immune complex may be responsible for the effect. Perhaps a simple experiment with blebbistatin can help. I am not sure if blebbistatin works on Xenopus myosin II. If not, they can first supplement the extract with human myosin II (after depletion of Xenopus myosin II) followed by blebbistatin addition. Such an experiment of humanized-Xenopus extract can answer all concerns mentioned in this point*.

To address this important point regarding the role of myosin in cortical symmetry breaking, we preformed additional experiments in which we added purified myosin II to myosin-depleted extracts (Figure 3 in the revised manuscript). These experiments show that while no symmetry breaking is observed in myosin-depleted extracts, addition of purified myosin II restores symmetry breaking in a dose-dependent manner. As noted by the reviewer, myosin immunodepletion could remove additional factors from the extract. However, the fact that we can restore symmetry breaking by adding purified myosin indicates that myosin is the essential factor for cortical symmetry breaking. We thank the reviewer for this excellent suggestion, and believe that the revised paper is significantly improved by substantiating this point, which is a central to our work.

Inhibition of myosin with blebbistatin in our system is not feasible. First, small molecule inhibitors like blebbistatin are known to be significantly less effective in cell extracts (Fields, Mitchison et al. Methods in Enzymology, in press), with inhibition typically requiring 10-1000 fold higher doses in comparison to cell culture. Second, cell-permeable small molecule inhibitors are typically hydrophobic and thus partition mostly into the oil phase in our emulsion system. Indeed, when we tried to use blebbistatin in our system we found that it partitions primarily into the oil phase (evident by its yellowish color which colors the oil phase) and the residual amounts of blebbistatin in the aqueous phase had no detectable effect on myosin function. Thus while myosin inhibition with blebbistatin would in principle be a simple and informative experiment, it is not feasible in our emulsion system. However, given that we were able to show that symmetry breaking does not occur in myosin depleted extracts, and is restored by add-back of purified myosin, we believe that the essential role of myosin in cortical symmetry breaking is already well-substantiated by our results.

Reviewer 2:

*1) A key conclusion of the paper is that myosin contractility can drive symmetry breaking in an actin cortex reconstituted in a droplet. This is a very interesting finding. However, the conclusion relies on a single experiment: the immunodepletion of myosin II-A heavy chain in the extracts. It is unclear how much myosin is really depleted using this method, other myosins are likely to be present in the extract, and there is no experimental evidence that myosin really localizes to the cortex in this system. To truly conclude on the importance of contractility, it seems essential to use some alternative method to test the functional importance of myosin activity. Is it possible to use drugs (such as blebbistatin) in this system? Alternatively, could the localization and dynamics of myosin be imaged*
*during symmetry breaking?*

In response to the reviewer’s concerns, we have added new experiments that provide a direct test for the functional importance of myosin activity. As detailed in the response to Reviewer 1, we performed experiments in which we added purified myosin II to myosin-depleted extracts and showed that symmetry breaking is restored upon addition of the purified myosin. Thus we now provide direct proof that myosin II is essential for symmetry breaking: symmetry breaking does not occur in myosin-depleted extracts, and is restored following add back of purified myosin

*2) The authors allude several times to a dynamic cortex, stating that actin polymerization and depolymerization are important for symmetry breaking. However, this is not investigated directly. FRAP experiments to quantitatively measure actin dynamics would help to better describe the behavior of this system and to support the authors' conclusions on the importance of actin dynamics*.

We have added FRAP experiments that directly illustrate the dynamic nature of our artificial cortices (Figure 1, Videos 1 and 3) to address this point. We observe the recovery of the actin network in the photobleached region of the cortex within minutes. Similar recovery is seen both in symmetric cortices (Video 1) and within the actin cap in cortices that had undergone symmetry breaking (Video 3).

*3) It is puzzling that droplet deformation is observed in only 10 % of the polarized droplets. The authors speculate that this might be due to the relatively high tension of the droplet interface, which they estimate at 700 pN/μm. Could they give details on how this tension was measured? This tension is indeed rather high, but not that much higher than tension measured by the same group in migrating keratocytes (Lieber et al. 2013). Was any deformation observed in the droplets at higher temperature, where polarization was induced by addition of crosslinkers? One would expect that interfacial tension would be lower at a higher temperature*.

The surface tension in the system was measured using the pendant-drop assay. An extract containing droplet was injected into an oil cuvette, and the droplet was imaged from the side to determine its shape. The shape of the droplet, which is deformed from a spherical shape due to gravity, was analyzed, providing a direct estimate of the surface tension. We have added a description of the surface tension measurements using the pendant-drop assay to the Methods section in the revised manuscript. The surface tension in our system is much lower than a bare water-oil interface, and is indeed close to the upper limits of membrane tension we measured in motile keratocytes, but it is still quite high relative to most other cell types.

Note that the contractile forces within the cortical network should be primarily tangential to the interface. The observed deformation must result from a net force perpendicular to the interface, which could arise from a small perpendicular component of the contractile forces in the cortex or from protrusive forces due to actin polymerization at the interface.

We observed deformed emulsions in experiments performed at higher temperatures in which crosslinkers were added to induce symmetry breaking. Given the relatively low overall incidence of deformed emulsions, we do not have sufficient statistics to determine if the incidence of deformed emulsions increased under these conditions. In any case this effect is expected to be small since the temperature-dependent change in surface tension in our system (between 20°C and 30°C) is rather small.

Reviewer 3:

*Specific*
*comments:*

*2) Water-oil emulsions have been previously used as biomimetic system for the study of cell-cell adhesion (Biomimetic emulsions reveal the effect of mechanical forces on cell-cell adhesion, 2012, Pontani et al. PNAS). This study should be cited as precedent for the technique and could be included in the Discussion regarding potential extension for artificial cells*.

We have added a reference to this paper.

*3) The system should allow comparably simple analysis of cap formation. The authors should therefore provide more information about the exact time course of cortex polarization: parameters include persistence of clusters, rate of movement (not only the initial flow speeds). In particular show formation of initial clusters with better temporal resolution*.

We have added additional characterization of the dynamics of the system. As noted in the response to Reviewer 2, we have added the results of FRAP experiments characterizing the rate of actin turnover in the cortex (Figure 1, Videos 1 and 3). We have also added data of the overall rate of movement of the cap (Figure 2—figure supplement 3), as well as data showing the behavior of the system over longer time scales (Figure 2—figure supplement 3). Since it is not possible to image the system over time in 3D due to phototoxicity effects, it is difficult to assess the persistence of clusters in a reliable fashion and discriminate between disappearance of clusters and movement out of the focal plane.

*4) Use FRAP to characterize lateral mobility and turnover of the cortex (actin and ActA)*.

As noted in the response to Reviewer 2, we have added FRAP analysis of the actin turnover in the cortex. The actin signal recovers within minutes after bleaching a region in the cortex, both in symmetric cortices and polar actin caps (Figure 1, Videos 1 and 3). The recovery appears rather homogenously within the bleached region. We have also added the results of FRAP experiments of the ActA at the interface (Figure 2—figure supplement 6).

*6) In*
Video 2
*it appears like cytosolic patches are moving in coordination with the cortical cap. It is possible that the actin network also extends into the extract and actually represents a 3D meshwork. This would be closer to the previous studies on simple reconstituted actin networks. Can EM images be obtained from fixed slices*
*to test this?*

It is very likely that a sparse network of actin filaments is present within the bulk of the emulsions. Individual actin filaments will not be visible in our system, but we do occasionally observe elongated actin structures within the emulsions (must likely several filaments or bundles). For example, such internal networks are evident in Video 4 and appear to move in coordination with the contraction of the cortex. This behavior is very reminiscent of in vivo observations in developing embryo, where similarly cortical flows are observed mainly at the interface, but appear to drive contraction of a much sparser actin network within the cytoplasm. As mentioned earlier, sample preparation for electron microscopy is challenging with the emulsion system. In particular, it is not possible to chemically fix the cortices within the emulsions.

*7) What is the nature of internal actin/ActA structures in deformed drops? Does actin cover small lipid droplets? Can a lipid dye be used*
*as counter stain?*

After long incubation times (>40-50 min), assymetric cortices usually develop internal actin/ActA structures characterized by aggregates of discrete ActA/actin puncta near the cap (regardless of the appearance of deformation in the droplets shape; see e.g., Figure 2—figure supplement 3). These structures are not observed in homogenous cortices after similar incubation times. Similarly, these structures are not seen during cortex formation at shorter incubation times (< 30 min). However, discrete puncta of ActA are always present in the extract. This is not surprising, since the extracts are known to contain lipids and small vesicles and the amphiphilic ActA will localize to these sites. Since the aggregates appear only in emulsions which display contractile behavior, we believe that these structures arise due to the contraction of a sparse network of actin filaments nucleated around these small vesicles or present in the bulk of the emulsions. This network probably pulls small ActA coated vesicles with it as it contracts. Using a lipid dye will probably not be very informative since it will likely exhibit some colocalization with the amphiphilic ActA molecules regardless of the nature of the internal structures observed

*8) Easy additional control experiments would be inhibition of myosin by blebbistatin (to show that myosin activity is needed) and depletion of specific crosslinkers from extracts. Why was only one concentration used for filamin? Show dose-response curves to better compare (maybe crosslinkers are not all equal). Can the published effect of pH on cortex organization (Bausch lab) be reproduced*
*in the new system?*

As discussed above, the use of blebbistatin on crude extracts and in particular within water-in-oil emulsions is not possible. Therefore, we tested the role of myosin using depletion experiments and supported our conclusion by adding purified myosin to restore the symmetry breaking.

We are aware that different actin crosslinkers are substantially different. The experiments in which crosslinkers were added to extracts were done to show that the reason that spontaneous symmetry breaking is not observed at higher temperature is the lack of connectivity in the network, rather than insufficient motor activity. The point of using both α-actinin and filamin, was to demonstrate that the symmetry breaking can be induced by different types of crosslinkers, emphasizing the importance of network connectivity, rather than the detailed characteristics of different crosslinking molecules. We believe that careful dose-dependent characterization of different crosslinkers and analysis of the behavior of the system at different pH values (as beautifully done in the work by Kohler et al. from the Bausch lab) is beyond the scope of the current manuscript.

[Editors’ note: the author responses to the re-review follow.]

Reviewer 1:

*In summary, I would in principle support publication provided that the authors include a paragraph in their Discussion that clearly states limitations of the system (difficult fixation, phototoxicity, problems with lipophilic drugs). In addition, possible approaches to gain new fundamental insights into the symmetry breaking mechanism using the emulsion system should be identified and discussed*.

We thank the reviewer for this suggestion and have added a section in the Discussion that clearly addresses these important points.

Reviewer 2:

*The authors did a good job at addressing the reviewers' comments and the paper now presents a thorough description of this very interesting experimental system. In particular, the myosin addition experiment indeed demonstrates that myosin drives symmetry breaking in this system. Are the patches formed upon addition of large amounts of myosin (*Figure 3*, right panel) dynamic, like in the* C. elegans *cortex, or static*?

We thank the reviewer for the appreciation of our work. The patches that form at high myosin concentrations are dynamic and exhibit jittery movement. We mention this in the revised manuscript. A thorough analysis of the dynamics of the cortices under these conditions and the very relevant comparison to the *C. elegans* cortex is beyond the scope of this manuscript and will be analyzed in detail in the future.

*I have some concerns about the photobleaching experiments, which are important for the message of the paper: in the absence of quantification, it is unclear how representative they are. Some quantification (e.g., average recovery time) and an example of recovery curve for the cap and the uniform cortex, should be provided*.

We have added a more thorough analysis of the FRAP experiments to the revised manuscript. Cortical recovery after photobleaching was rather consistent between different cortices, and the examples shown in Figure 1 and Videos 1 and 3 are representative. We now provide histograms of the half time for recovery at 30 °C and at 20 °C, and show the time evolution of the intensity during the recovery process to demonstrate this (Figure 2—figure supplement 3). The recovery at 30°C and at 20°C appeared similar.

The cortices typically exhibited only partial recovery, usually returning to ∼50 % of their initial intensity. The partial recovery appears to be a result of photodamage to the actin network, which is known in similar systems (e.g., Simon et al. Biophysical Journal 1988; Niedermayer et al*.* PNAS 2012). The partial recovery was not due to an immobile actin population at the interface, as repeated bleaching at the same cortical region did not result in full recovery. Moreover, when we considerably increased the FRAP illumination dosage we did not observe any cortical recovery.

*Also, did photobleaching never induce symmetry breaking? Photobleaching of actin networks* in vitro *has been reported previously to induce symmetry breaking (possibly by photo-induced network disruption, e.g., van der Gucht et al. 2006)*.

The photobleaching did not typically induce symmetry breaking in our artificial cortices. At 30°C the system is not contractile, so even if the cortices were locally disrupted we would not expect large-scale symmetry breaking. At 20°C the cortices were typically non-symmetric to begin with.